# Mobilome of the *Rhus* Gall Aphid *Schlechtendalia chinensis* Provides Insight into TE Insertion-Related Inactivation of Functional Genes

**DOI:** 10.3390/ijms232415967

**Published:** 2022-12-15

**Authors:** Aftab Ahmad, Zhumei Ren

**Affiliations:** School of Life Science, Shanxi University, Taiyuan 030006, China

**Keywords:** *Schlechtendalia chinensis*, transposable elements, domestication, transposase-derived protein, insertional inactivation

## Abstract

Transposable elements (TEs) comprise a considerable proportion of insect genomic DNA; how they contribute to genome structure and organization is still poorly understood. Here, we present an analysis of the TE repertoire in the chromosome-level genome assembly of *Rhus* gall aphid *Schlechtendalia chinensis*. The TE fractions are composed of at least 32 different superfamilies and many TEs from different families were transcriptionally active in the *S. chinensis* genome. Furthermore, different types of transposase-derived proteins were also found in the *S. chinensis* genome. We also provide insight into the TEs related insertional inactivation, and exogenization of TEs in functional genes. We considered that the presence of TE fragments in the introns of functional genes could impact the activity of functional genes, and a large number of TE fragments in introns could lead to the indirect inactivation of functional genes. The present study will be beneficial in understanding the role and impact of TEs in genomic evolution of their hosts.

## 1. Introduction

The genomes of all insects vary in size, ranging from 99 Mb of *Belgica antartica* [1] to 8.55 Gb of desert locust *Schistocerca gregaria*, an almost 100-fold larger genome than the former [2]. However, a considerable portion of insect genomes is occupied by repetitive DNA, which primarily consists of transposable elements (TEs), also called genomic parasites, and autonomous TEs encode the genes, enabling them to move and replicate in the genome, e.g., transposase [3]. TEs are known as junk or selfish DNA fragments that can translocate from one part of a genome to another, and often account for a large proportion of the eukaryote genomes, including insects [4]. Due to their mobile nature, TEs replicate and insert into new positions in the host genome, contributing to a large proportion of the host genome size [3]. Because TEs move in host genomes, insert into novel sites, shape and induce changes in the landscape of coding genes and non-coding DNA fractions, they are considered the most potent internal sources of genotypic changes [5,6].

TEs are divided into two groups depending on their structural makeup and transposition process. The Class I elements, commonly known as retrotransposons (RTs), are known to transpose via an RNA intermediary and self-replicate after transposition. Class II elements, often known as DNA transposons, can hop between genomes through a straightforward “cut-and-paste” technique. Furthermore, subclasses, orders, and superfamilies of TEs can indeed be derived from the two classes [3]. RTs are typically the most frequent repeated elements; after being transposed, they can increase their copy numbers, particularly in plant genomes [7]. However, the number and abundance of RTs differ throughout insect genomes. Long terminal repeat retrotransposons (LTR-RTs), for example, include four superfamilies, namely Copia, Gypsy, Bel, and endogenous retroviruses, while non-LTR-RTs have almost 30 superfamilies, including L1, R2, *Jockey*, and CRE [8]. Non-LTR-RTs nick target sites and reverse transcribe utilizing endonuclease, whereas LTR-RTs reverse transcribe the cDNA and integrate with the help of integrase [9,10]. Class II TEs translocate using the DDE transposase enzyme that excises and integrates TEs to new sites by the “cut and paste” mechanism. Approximately 20 superfamilies of Class II TEs use DDE transposase protein, i.e., Tc1/Mariner, Harbinger, EnSpm, and Mutator-like elements (MULE) [9,10]. Superfamily Crypton translocation requires tyrosine recombinase and circular DNA intermediates, whereas Helitron transposes from a double-stranded circular DNA and integrates into the new site by a “peel and paste” method [10,11].

When TEs translocate in genomes, they are well recognized for producing genetic changes and can impact gene expression directly or indirectly by affecting the expression of nearby genes [10]. Because the unrestricted activity of TEs is usually harmful, the host organism has multiple defense systems in place to limit TE mobility at various points of their transposition cycle [11,12]. Activating TEs in response to stress may result in random genetic alterations that result in variances and, on rare occasions, genotypes that are better adapted to the relevant environmental stress. There is some evidence that stress-induced transpositions and alterations or changes in gene expression may be advantageous for stress adaption [13]. TEs are dispersed throughout the genomes, and many studies reported TEs to contribute to the evolution of the gene regulatory network [14,15] and provide transcriptional starting sites for the neighboring genes [15]. TEs could have a negative impact on genomes and gene expression in response to the activation due to stress [13].

These mobile sequences (TEs) are known to play roles in various evolutionary scenarios, such as gene creation by domestication and gene regulation, chromosomal rearrangement, and genome evolution [16,17,18,19]. In addition, they can have a harmful effect on neighboring genes, and TE insertions into the exon could lead to the inactivation of neighboring genes [20].

The *Rhus* gall aphid *Schlechtendalia chinensis* (Hemiptera: Aphididae: Eriosomatinae: Fordini) induces gall on the leaves or branches of its primary host plant *Rhus chinensis* and lives inside it for three generations of their life cycle [21]. The galls produced by this aphid are rich in tannins and widely used in the medicinal, chemical, and food industries, and have high economic value [22]. Galling aphids have unique evolutionary adaptations and may serve as a suitable model for studying insect–plant interaction on their unique behavioral and ecological phenomenon and coevolution [23,24]. Thus, identifying and annotating TEs of the genome of galling aphids will provide insights into the diversification of TEs in aphids and will lay a foundation for investigating the impact of TE insertions on neighboring gene activity and genome evolution.

In this study, we sequenced and assembled a detailed and advanced chromosome-level genome and used different methods to identify and annotate the TEs landscape in the genome of *Schlechtendalia chinensis*. We also focused on the insertions of TEs in different sets of functional genes, analyzed and profiled the transcriptionally active TEs from the whole transcriptome of *S. chinensis*. Furthermore, we analyzed the insertion of TEs into the functional genes from three families and their impact on the expression of selected functional genes. This study will present the first detailed report on the TEs landscape, and characterization in the genome of *Rhus* gall aphid.

## 2. Results

### 2.1. TE Identification and Annotation

We analyzed a high-quality 344.59 Mb genome assembly of *Schlechtendalia chinensis* with 91.71% of the assembled sequences (315.55 Mb) anchored into 13 chromosomes. We detected a total of 308,196 TEs in the genome of *Schlechtendalia chinensis* using signature and homology-based strategies. The whole TEs dataset included both complete and incomplete copies and covered 89,735,022 bp, accounting for 26.04% of the total genome of *S. chinensis*. The full-length TEs dataset consists of 7743 elements, including LTR-RTs, non-LTR-RTs, and DNA transposons, accounting for approximately 7% of the whole genome. Among the LTR-RTs, Gypsy and Pao have a more significant number of full-length copies, while LINE has more full-length copies than non-LTR-RTs. The total number, full-length copies number, classification, and percent contribution of TEs to the whole genome were also analyzed. All the TEs identified in this study were further analyzed and located on the chromosomes, and their density in 13 chromosomes can be seen in the genome of *S. chinensis* (Figure 1a).

### 2.2. Class I Retrotransposons

Retrotransposons are the main focus of many research studies, and they contribute to a significantly higher proportion of species genomes. All the RTs in *S. chinensis* were harvested and identified using different strategies and TEs databases (see Section 4) (Table 1). We identified 96,053 RTs, which covered 33,038,958 bp and accounted for 9.59% of the genome, among which a total of 2777 complete and nearly complete elements accounted for approximately 3.12% of the entire genome. We classified all the detected RTs into two major orders (LTR-RTs and non-LTR-RTs) and ten superfamilies based on homology, shared signatures, and domain with the known families using TEs libraries (Figure 1b).

LTR-RTs isolated from the genome *S. chinensis* belonged to diverged lineages and were initially classified into five known superfamilies and one unknown lineage (Table 1). LTR-RTs covered 15,428,927 bp, accounting for 4.48% of the total genome. Gypsy is the most abundant superfamily among the known superfamilies and contains 18,239 elements, including 2002 full-length and partially full-length copies, with sizes ranging from 1133 to 12,751 bp. Pao is the second most abundant superfamily containing 2624 elements and 179 complete copies with a size range from 1324 to 13,081 bp. The third most abundant superfamily is Copia, which includes 1796 elements and 100 complete copies ranging from 1100 to 10,982 bp. We used LTR-retriever to harvest copies with intact long terminal repeats and target site duplication (TSD) signature in LTR-RTs. We found 661 complete copies with TSD and flanking repeats at both ends belonging to the Gypsy, Pao, and Copia superfamilies of LTR-RTs (Appendix A). Among these, we identified 30 complete LTR-RTs with flanking terminal repeats and TSD at both sides that belong to unknown lineage due to lack of homology with known lineage-specific RT protein domains.

Phylogenetic analysis of the representative full-length LTR-RTs of the Gypsy, Pao, and Copia superfamilies was performed by a homology-based approach. A phylogenetic tree of 309 full-length representative sequences of the Gypsy, Pao, and Copia superfamily was constructed, classifying them into three, three, and four known clades, respectively (Figure 2a). An unknown lineage of 10 elements in Gypsy, 30 in Pao, and 10 in Copia was defined because there are not enough similarities between the RT protein and recognized lineage-specific domains.

In addition, a small number of RTs from the ERV superfamily were identified with a maximum fragment length of 1502 bp. A few incomplete copies of Cassandra and Caulimovirus superfamilies were isolated, and a significant number of unknown LTR-RTs were also isolated with missing conserved domains and motifs, covering 0.88% of the whole genome.

A total of 54,029 non-LTR RTs were identified with 16,374,069 bp in length, occupying 5.11% of the *S. chinensis* whole genome (Figure 1b, Table 1). LINE superfamily is the most widely spread non-LTR-RTs, containing 46,644 elements and covering 16,342,622 bp (4.74%) of the whole genome. We harvested 276 full-length LINE-RTs, whose sizes range from 4000 bp to 16,000 bp with a median length of 7026 bp. Phylogenetic analysis of the 219 representative sequences of the LINE superfamily was performed to classify them into 18 known sub-lineages of seven major clades as well as to four unknown lineages due to vague similarities to the known lineage-specific domain of LINE-RT (Figure 2b). A considerable amount of SINE RTs are also dispersed in the *S. chinensis* genome with a total of 6852 and covering 1,234,961 bp (0.36%) of the whole genome. The size of the full-length SINE elements ranges from 600 to 1050 bp, with a median length of 724 bp. A few hundred sequences from the DIRS superfamily were also isolated, covering 26,185 bp of the whole genome, with a maximum sequence length of 244 bp. At the same time, a few deleted copies of Viper non-LTR RTs were also identified. The detailed information, location, numbers, and classification of all the Class I TEs are listed in Appendix A.

### 2.3. Class II DNA Transposons

Class II TEs covered the most proportion of the *S. chinensis* genome. We identified 212,638 DNA transposons, including both full-length and incomplete copies, which covered 56,696,064 bp and accounted for 16.45% of the total genome of *S. chinensis*. We isolated 4966 full-length sequences of DNA transposons with the sequence length ranging from 1200 bp to 21,213 bp (Appendix A). Approximately 48.1% of the DNA transposons belonged to the TIR order, and 45.7% of the sequences were classified as unknown due to lake of similarity with already known TEs.

All the TEs from the TIR order were further classified into 17 superfamilies, in which hAT and Tc1/Mariner superfamilies were the most abundant. Other identified elements belonged to the CACTA, MULE, PIF-Harbinger, PiggyBac, P elements, Ginger, Zisupton, Novosib, Kolobok, Sola, IS3EU, Merlin, Dada, Zator, and Academ superfamilies. A total of 1929 incomplete elements belonging to the Crypton superfamily of order tyrosine recombinase were also retrieved. The present result indicates that DNA transposons in the *S. chinensis* genome are abundant and widely spread, which covered more of the genome than RTs (Table 1 and Figure 1b).

The hAT superfamily of DNA transposons, one of the widely spread superfamilies in eukaryotes, was named from three of its member transposons: hobo element, Activator element, and Tam element. The analysis of the *S. chinensis* genome revealed that hAT is the most dispersed and abundant superfamily of DNA transposons. We identified 47,622 hAT elements in the *S. chinensis* genome, covering 12,144,219 bp and accounting for 3.52% of the whole genome (Table 1). Among all the hAT elements, 990 sequences were full-length, ranging from 2003 to 12,752 bp. Structural analysis of the full-length copies indicates that many sequences have the intact open reading frame (ORF) and are potentially active. A total of 20 hAT elements have intact open reading frame TIRs and are considered potentially active (Appendix A).

Phylogenetic analysis of the 200 representative sequences of hAT elements from the dataset was performed by constructing an ML tree. All the hAT sequences were clustered into six known families, i.e., Tip100, Tag1, Activator, hATx, hAT19, and Charlie (Figure 3a). The naming and classification were validated using Dfam database and online library, and all the families were named following Dfam classification. The current analysis shows that the hAT superfamily is the most abundant and widely spread in the *S. chinensis* genome. The presence of complete ORF and the potential ability to transpose within the genome may be one of the reasons that the hAT superfamily is the most abundant DNA TE in the *S. chinensis* genome. It could play a crucial role in shaping and evolving the *S. chinensis* genome.

The Tc1/Mariner superfamily is one of the well-characterized groups of DNA transposons and is considered the most widely spread superfamilies of DNA TEs. Tc1, Mariner, and IS630 were supposed to represent different families but later combined as one IS630/Tc1/Mariner superfamily based on homology and a shared transposition mechanism. We identified all the Tc1/Mariner elements and found them to be the second most widespread DNA TEs after the hAT superfamily in the *S. chinensis* genome.

We recovered a total of 16,200 DNA TEs belonging to the Tc1/Mariner superfamily, which covered 3,923,428 bp and accounted for 1.14% of the whole genome of *S. chinensis*. Among the total elements, only 571, ranging from 1180 to 10,242 bp, were identified to be full-length, having flanked TIRs and internal transposase domain. All the full-length copies were analyzed for the internal transposase ORF, and 65 of the elements had intact ORF, conserved catalytic and helix-turn-helix HTH domains, and were considered potentially active. Phylogenetic analysis of the full-length copies of Tc1/Mariner superfamily was performed to present the evolutionary relationships of these TEs in the *S. chinensis* genome. We constructed an ML tree using 300 TEs belonging to the Tc1/Mariner superfamily in *S. chinensis* along with the already known families of superfamily Tc1/Mariner from GenBank. Interestingly, all the TEs were grouped into four major families, i.e., Mariner, Tc1, DD40D/Visitor, and M44 family (Figure 3b). TEs from the three families, i.e., Tc1, Visitor, and M44, were identified in the *S. chinensis* for the first time, while the Visitor and M44 families are newly discovered and we reported them in gall aphid for the first time in this study. The TEs from the recently reported DD40D/Visitor family were confirmed by analyzing its DDE catalytic domain and found to be more abundant than mariner elements in the genome of *S. chinensis*. The M44 and Tc1 families were identified and classified based on homology, following Dfam classification. The Mariner family is the most diverged among the Tc1/Mariner superfamily and is further classified into three known families (Mauritiana, Irritans, and Drosophila), and one un-defined subfamily due to a lack of homology with the known subfamilies (Figure 3b).

Structural analysis of the full-length copies of the DD40D/Visitor family reveals that many sequences from the family have intact ORF for transposase and TIRs flanking at both ends, which could be potentially active. A small proportion of sequences were clustered in an unknown group due to a lack of conserved domains which were similarities with the known families. All the full-length sequences from the Tc1/Mariner superfamily used in the phylogenetic analysis are given in Appendix A.

The elements of other significant superfamilies of DNA TEs, i.e., CACTA, P-elements, PiggyBac, Merlin, and MULE, were also identified, acquiring a considerable proportion of *S. chinensis* genomes with few potentially active copies. PIF/Harbinger, Novosib, and Ginger superfamilies contain full-length or slightly truncated copies. The remaining superfamilies, including Academ, Dada, IS3EU, Kolobok, Sola, Zator, and Zisupton, have a small number of deleted and truncated copies present in the *S. chinensis* genome. The MITE superfamily, well-characterized in other metazoans, contains only 31 elements in the *S. chinensis* genome (Table 1). These superfamilies have been annotated and classified automatically using reference libraries of Repbase and Dfam (Appendix A).

### 2.4. Subclass 2 (Helitron and Maverick) DNA TEs

Subclass 2 contains two groups of DNA TEs, and they are placed into separate classes based on their transposition mechanism. These TEs are also known as rolling circle transposons because of their replication mechanism. The Helitron and Maverick TEs are well characterized and can be found in diverse species from protists to higher animals and plants. We carried out a systematic search for the mining and identification of Helitron and Maverick in the genome of *S. chinensis*. We found 7503 copies of Helitron with 2,623,664 bp in total length, contributing to 0.76% of the whole genome of *S. chinensis*. Among the copies, 158 were complete and partially complete, ranging from 2010 to 4223 bp long, inserted in 12 chromosomes. Structural analysis of the full-length copies revealed that only ten sequences had all the ORFs required for the process of transposition. The size and the distribution of complete sequences of Helitron are shown in Appendix A.

The Maverick TEs, also called Polintons, are considerably longer than other DNA TEs and are a complex family of TEs. We identified 3899 TEs from the Maverick superfamily covering 4,729,237 bp and accounting for 1.37% of the whole genome of *S. chinensis*. A total of 218 full-length or slightly truncated sequences were recovered with a size range from 5008 to 14,868 bp (Appendix A). The complete sequences were analyzed for potentially active Maverick sequences and we did not find any sequence with all 10 intact ORFs required for transposition. The Maverick superfamily has the most complex TEs, and due to their large size and significant copies, they may play a crucial role in *S. chinensis* genomics. Our present results provide the basis for further research.

In summary, we present the dynamics and evolution of all the TEs in the *S. chinensis* genome and describe the diversity and expansion of the significant families of TEs in the *S. chinensis* genome.

### 2.5. Transcriptomics Analysis of the Potentially Active Copies

The TEs profile of the *S. chinensis* genome revealed the presence of a large number of full-length potentially active copies. These results prompted us to further investigate the expression of the TEs identified in the analysis. To this end, we mined the TEs transposase sequences in the whole transcriptome of *S. chinensis*. Surprisingly, we annotated hundreds of transposase sequences (complete and fragments) belonging to RNA and DNA-mediated TEs that showed their expression activity (Appendix A). The Class I RNA-mediated transposase sequences include Pao-derived transposase, the most abundant and expressed TE in the *S. chinensis* genome. Only two transcripts of Copia-derived transposase (incomplete) were found in the transcriptomic analysis. Transposase derived from non-LTR TEs such as LINE has not been actively expressed in the genome of *S. chinensis* (Table 2).

Transcripts belonging to DNA transposons were found to be the most abundant, and TEs from many superfamilies are transcriptionally active in the genome of *S. chinensis*. Mariner Mos-1-derived transposase was found abundantly in the transcriptomes, which indicates their high transcriptional activity in the *S. chinensis* genome. Mariner TE is one of the most abundant families in *S. chinensis*, and many potentially active copies were characterized in its genome. There are few potentially active copies of PiggyBac, Mutator, and hAT present in the *S. chinensis* genome. The transcriptomic result shows that they are transcriptionally very active and might be involved in the transposition of TEs. Further research will be required to analyze and confirm the activity and transposition of these elements. A small number of transposase sequences derived from subclass 2 of DNA TEs such as Helitron were also found in transcriptome analyses (Appendix A). We did not find any transposase sequence derived from Maverick. In conclusion of the present analysis, we mined the whole transcriptome of *S. chinensis,* and reported several transcripts belonging to known TEs, indicating only their transcriptional activity. Further studies would be required to investigate and analyze the transposition activities of these TEs in the *S. chinensis* genome.

### 2.6. Domesticated TE/Transposon-Derived Proteins in the Genome of S. chinensis

TEs have been considered as junk and selfish elements invading the genome of living organisms, from prokaryotes to eukaryotes. Several multigenic families have been derived from different molecular domestication events of TEs. We also analyzed the annotated genomes of *S. chinensis* to detect TE-derived proteins. We found many protein families derived from transposase or chimeric proteins that contained at least one of the transposase domains. We annotated all the protein sequences containing transposase domains derived from RT-TEs and DNA TEs. We found one protein family associated with zinc fingers (znf), nucleic acid binding activity derived from transposons X (an element derived from the LINE element of RT-TEs). We did not find any other protein family derived from RT-TEs. We identified many protein families derived from the DNA TEs transposase, such as Mariner Mos1, hAT, piggyBac, Harbinger, and Tigger/Tc5.

The SETMAR protein produced by the fusion of Mariner transposase and Methnase protein was the first domesticated protein reported in primates. The Mos1 transposase, beingpart of SETMAR, is abundantly found in the genome of *S. chinensis*, but did not have the Methnase part. Other DNA transposase-derived proteins include Mariner Mos1 transposase-derived GVQW3 protein (HTH-48 domain-containing), PiggyBac-derived chimeric ERCC6-PGBD3 protein, Harbinger transposase-derived HARBI1 nuclease protein, Tigger transposase-derived protein (containing Homeobox-like, and HTH CENPB-type DNA-binding domains), metastasis-associated protein MTA, hAT-derived Zinc finger MYM-type protein, zinc finger BED domain-containing protein, general transcription factor II-I repeat domain-containing protein 2 (all containing hAT family C-terminal dimerization region), and hAT-derived TPR_REGION domain-containing protein. We also found many uncharacterized protein sequences derived from transposons with unknown functions (Figure 4).

To confirm the evolutionary relationships and understand the molecular domestication of transposon-derived proteins found in *S. chinensis*, we carried out their phylogenetic analysis. We performed BlastP searches in NCBI databases for all the transposon-derived proteins and found them in many organisms, from insects to higher vertebrates, including humans. The ML phylogenetic tree of all the sequences indicated that the transposase-derived proteins presented in the *S. chinensis* genome diverge among different organisms (Figure 5, Appendix A). The presence of these proteins in different lineages further provides insight into their ancient molecular domestication in their common ancestor during evolution.

All the TE-derived sequences were searched with BlastP in multiple proteindatabases and annotated accordingly. We also carried out KEGG and GO enrichment analysis for all the transposon-derived protein sequences, which show they are involved in different cellular and biological functions (Appendix A). NCBI BlastP search reveals that all the sequences have been present in other aphids and insects, reflecting their ancient molecular domestication during evolution.

### 2.7. TE Insertions into Functional Genes

It is widely thought that TEs are the major source of mutations of host functional genes. We analyzed the nucleotide sequences and structures of several gene families to explore the impact and insertions of TEs into functional genes. A total of 101 gene sequences from three families, i.e., Cytochrome P450, Carboxylesterase, and Trypsin (Appendix A), were masked with the Censor tool using the Repbase TEs library with default parameters. Initially, a total of 2065 regions were masked in all three gene families with a size ranging from 33 bp to 1055 bp. The high sensitivity of the Censor can also lead to the masking of small GC and AT-rich regions within introns and exons, and could lead to false positives (low scoring results). To avoid false positives, we considered fragments with length ≥100 bp as inserted TEs, and the fragments with a size less than 100 bp were excluded from the analysis (Appendix A).

Cytochrome P450 monooxygenase gene family of *S. chinensis* consists of 36 genes, of which one gene is a probable pseudogene. A total of 186 TE fragments ranging from 100 to 1055 bp were found to be inserted in 29 out of 36 sequences, accounting for 31,777 bp. A total of six fragments with sizes ranging from 108 to 151 bp were inserted in the exons without interrupting the coding ability, while the remaining insertions were intronic. Interestingly, the 22,884 bp pseudogene is the largest in the Cytochrome P450 family and has 24 insertions of TE fragments, with the largest fragment having a length of 1055 bp. We speculated that a large number of TE insertions in the pseudogene might cause its inactivation. We analyzed the gene structure and found that all the TEs inserted in the introns except one 127 bp fragment in the exon region of the Cytochrome pseudogene and did not directly interrupt the coding ability. From the transcriptome data analysis of *S. chinensis*, the gene’s protein product (pseudogene) is 152 amino acids long. The actual gene product was supposed to be a 504-amino-acid peptide.

To investigate whether the TEs inserted in this gene have a role in the inactivation of the gene into pseudogene, we analyzed the first and the longest intron of the gene. The intron is 21,135 bp long, having 21 TE fragments out of 24 TEs in the whole gene. We used the Spliceator tool online [25] (for splicing sites predictions) to analyze the splicing sites. Interestingly, we found several splicing sites (Donor and Activator) within the TE fragments present in the introns (Figure 6a and Table 3), which could result in abnormal splicing of mRNA and the inactivation of the genes. Splicing error due to TE fragments could lead one or more TEs to be the part of mature mRNA containing several stop codons. In conclusion of the above analysis, TE insertion in introns could lead to abnormal splicing, splicing mutation, and the inactivation of functional genes.

We also analyzed the other two gene families following the same procedure and found 195 TE fragments inserted in 19 of the total 20 genes of the carboxylesterase family. All the fragments were inserted in the intronic regions except three fragments (111 to 151 bp long) in the exons without disrupting the coding ability of the genes. All the genes from this family are transcriptionally active, and there is no observed direct impact of TEs on the structure and activity of the genes (Table 3).

Trypsin enzymes are essential for digestion and are functionally important. The Trypsin gene family in *S. chinensis* consists of 44 genes with variable lengths and structures. We analyzed the Trypsin gene family for TE insertions and detected 151 fragments of varying length range from 111 to 919 bp (Table 3). A total of 18 TE fragments were inserted in the exons of the genes without interrupting the coding ability of the gene. Surprisingly, the largest TE fragment, 919 bp in this family, was found in the exon of a gene named (Schi07G002340). The inserted TE belongs to hAT transposons of DNA TEs. This large TE fragment size in the gene’s coding region indicates ancient insertions of these TEs in the genes, which were removed from the genomes by purifying selection, and some part of the TEs has been domesticated (exogenized) (Figure 6b).

The above analyses and the presence of many TEs in the functional genes provide insights into the importance of TEs in host genomes. Further research would need to investigate the impact of TEs on the functional genes of the host.

## 3. Discussion

TEs contribute to a significant proportion of species genomes, and the variations in genome size of eukaryotes, including insects [26], are generally expected from the changes in the number of transposable elements. We used combined annotation strategies and TEs mining tools to delineate the TEs repertoire in the 13 chromosomes of *Schlechtendalia chinensis*. We searched the TEs in the genome of *S. chinensis* with high-quality TEs databanks, and the TEs represent 26.04% of the *S. chinensis* whole genome. The TEs content in *S. chinensis* genome assembly is significantly lower than those described in the non-galling aphid *Acyrthosiphon pisum* genome (38%) (IAGC, 2010) [27], and *Aphis glycine* (26.9%) [28]. In contrast, the TE contents of *S. chinensis* is significantly greater than those described in the *Diuraphis noxia* genome (15.31%) [29]. These variations in TE content in genomes could be attributable to extinction or variance in TE propagation and the influence of random population effects [30]. In comparison to prior research, we used more TE detection tools and conducted a more detailed analysis, encompassing even the most minor TE repeats.

The comparison of genome proportions of TEs with other aphids reveals that TIR elements from Class II TEs are the most abundant in the genome of *S. chinensis*, followed by the SINE repeats of Class I, which is incongruous with the previous report of the *D. noxia* genome [29]. The repetitive sequences in the genome *S. chinensis* are well annotated and classified in our present study. The abundance of TEs from unknown lineages is approximately 30.71 Mb, relatively less than 46.7 Mb of *A. glycine* and 32.2 Mb of *D. noxia* unclassified TEs. We presented a detail of TEs landscape and proliferation in the *S. chinensis* genome, which will be helpful for comparative genomic studies with any other galling or non-galling aphids in the future.

The RTs are widespread in the genome of *S. chinensis*, all containing major superfamilies. Gypsy is the most abundant, followed by ERV and Copia superfamilies, which were reported for the first time in galling aphids. In comparison, previous studies reported the presence of these elements in insects [31,32], including *Drosophila* [33], mosquitoes [34], and moths [35]. The LINE is the most abundant non-LTR-RT in the genome of *S. chinensis*, which was also previously reported in aphid *D. noxia* and many other insects [32].

The Class II elements are the most abundant and widely spread in the genome of *S. chinensis*, which is similar to other insects, i.e., Hessian fly *Mayetiola destructor* [36], mosquito *Anopheles gambiae* [37], jumping ant *Harpegnathos* [32], and the coleopterans *Anoplophora glabripennis* and *Tribolium castaneum* [38]. We here described most of the superfamilies among aphids in this study. The CACTA superfamily of Class II is one of the abundant superfamilies in *S. chinensis*. In contrast, this superfamily is not well characterized in other aphids, but widespread distribution has been reported in *Mayetola destructor* [36]. In insects, including aphids, the Tc1/Mariner superfamily is the most extensively distributed and well-characterized. Mariner/DD34D is the most widely dispersed transposon family in nature, and it has been extensively investigated and been found in Hexapods [32,39,40,41]. The Tc1/Mariner superfamily is classified into at least ten different families. Only the Mariner elements have been reported in *Rhus* gall aphids, including *S. chinensis*, that were also involved in horizontal transfer with other insects in our previous studies [42,43].

Transposable elements are abundant in all forms of life, but the active transposable elements are a hotspot for genomic research, as they can jump and actively multiply their numbers [44]. The transcriptomic analysis of the *S. chinensis* genome indicates that many TEs of different lineages are transcriptionally active. There are already several reports demonstrating the TE activity in other species in previous studies [10,45]. From the current study, it cannot be confirmed whether the transcriptionally active TEs are involved in active transposition or not, yet providing a basis for future research on these elements in aphids. Further research and experiments will be required to demonstrate the activity of transcriptionally active TEs.

It has been reported that TEs contributes to the functions and evolution of their host genomes through the donation of regulatory sequences that control the expression and regulation of nearby genes [45]. TEs also contributed to the protein sequences of host genomes by exogenization in the host gene [46], while domestication of transposase-derived protein has also been demonstrated in many studies [47,48,49]. We also identified many transposase-derived proteins in the genome of *S. chinensis*. Structural analysis of the proteins shows that they all derived from transposase, containing at least one domain similar to other functional proteins. Phylogenetic analysis revealed that these proteins were domesticated during the early course of species evolution and can be found in insects to humans. For instance, the RAG-1 protein, which has a role in V(D)J recombination, has been domesticated in higher vertebrates [47,50] and has also been found in *S. chinensis*, which indicates its ancient domestication during evolution. We did not find any new domesticated protein limited to *S. chinensis*.

TEs are believed to be a comprehensive source of mutations and genetic polymorphism due to their transposition. Many studies have described the role and impact of TEs on the host genome and functional genes [10,45], but no direct insertional mutagenesis has been reported yet. We analyzed and investigated three families of functional genes for the presence of TEs insertion. We found a large number of TE fragments inserted in the functional genes. All of the TE fragments were inserted in the intronic regions, except a small number found in the exon. TEs’ sequence in exons (exogenized) of functional genes without interrupting the coding ability has been reported in previous studies [51]. One gene from the Cytochrome P450s family lost its activity (become pseudogene) due to a large number of longer TE fragments in its intron. Many inactive TEs in the intron of the gene lead to splicing error by carrying abnormal splicing sites that could lead to abnormal mRNA and gene inactivation. The presence of a large number of TE fragments inserted in the Cytochrome P450 gene family has been reported in insects in a previous study, without any detail characterization and analysis [52]. We believe that the loss of activity by these processes might be related to TE insertions and provide some evidence of insertional inactivation of the functional gene by TEs. Our study also provides the first evidence of TEs’ direct insertions in the intron of functional genes. Further studies would be required to investigate other functional genes in different organisms.

## 4. Materials and Methods

The *Schlechtendalia chinensis* whole genome and transcriptome were sequenced in an ongoing lab project. This study used a high-quality chromosome-level genome assembly of *S. chinensis* to present the complete set of TEs. The *S. chinensis* genome sequence Sc2.0 version is deposited at NCBI (BioProject: PRJNA833747), and transcriptomic data are available at Ren Lab, School of Life Science, Shanxi University, Taiyuan, Shanxi, China.

### 4.1. Detection and Annotation of Transposable Elements

Transposable elements (TEs) were detected, identified, and annotated by combining homology-based and de novo approaches. Using RepeatModeler2, we first created a genome-wide de novo repeat library [53], which can run two de novo repeat searching scripts automatically, including RECON v1.08 [54] and RepeatScout [55]. Then, full-length long terminal repeat retrotransposons (LTR-RTs) were identified using LTRharvest with parameters (minlen: 100, maxlen: 40,000, mintsd: 4, maxtsd: 6, motif: TGCA, motifmis 1, similar 85, vic 10 -seed 20, seqids yes) and LTR_finder with parameters (D 40,000, d 100, L 9000, l 50, p 20, C -M 0.9) [56,57]. The high-quality intact LTR-RTs and non-redundant LTR library were then produced by LTR_retriever [58]. A non-redundant species-specific TE library was constructed by combining the de novo TE sequence library with the known Repbase v19.06, REXdb v3.0, and Dfam v3.2 databases [59,60,61]. Finally, TE sequences in the *S. chinensis* genome were identified and classified by homology search against the above libraries (Figure 7).

### 4.2. Phylogenetic Analysis

The classification of detected TE consensuses discovered in *S. chinensis* was inferred per superfamily using reference sequences from Repbase and Dfam. All the sequences in each analysis were aligned using MAFFT software [62] implemented in Geneious prime v.2022.1.1. The alignments were manually curated for dangling ends to produce an excellent phylogenetic tree. The tree for each superfamily was constructed using maximum likelihood using IQ software. The best model for each tree was predicted using the JModelTest [63,64]. All the phylogenetic trees were built with a bootstrap analysis of 1000 replicates. The trees were visualized and modified using Figtree software v1.4.4. All the TE sequences were classified into their respective lineages and families following the protocol proposed by Wicker et al. [3], and Dfam classification. TE sequences that did not classify in to known families following the Repbase and Dfam classification were named according to the RepeatModeler2 classification system [52]. Phylogenetic analysis of the transposase-derived protein sequences was carried out following the same procedure explained above.

### 4.3. Sequence Analysis and Identification

For the identification of full-length TEs, all the complete or nearly complete sequences were manually analyzed for their DIR in case of RT and terminal inverted repeats (TIRs) for the TIR order and target site duplications (TSD) accordingly. To find potentially active TEs, their protein-coding region was determined by the sequences using the ORF finder. For example, the Tc1/Mariner superfamily has only a single ORF coding for a transposase enzyme. So, the coding region was analyzed, and sequences having intact ORF and flanked by TIRs were considered potentially active. The conserved protein domain for all the potentially active copies was analyzed at least for one sequence from each group using the NCBI conserved domain search (https://www.ncbi.nlm.nih.gov/Structure/cdd/wrpsb.cgi,) with default parameters (accessed on 16 February 2021).

### 4.4. TE Insertions Analysis in the Functional Genes

The tree family of functional genes was masked using the Censor tool, implemented in Repbase online [65]. All the masked regions including both intron and exon were analyzed manually in all the genes. The exon of the gene was predicted using the ORF finder implemented in Geneious prime v.2022.1.1. The splicing sites in the pseudogene were predicted using the Spliceator tool online [25]). The figure was manually produced and labeled using PowerPoint, masking results.

Transposase-derived proteins were analyzed and annotated using public annotation databases such Pfam, Swissprot, NCBI CD search [66,67], and Motif search online (www.genome.jp/tools/motif/) with default parameter (accessed on 20 February 2021). The figure was manually produced and labeled using PowerPoint.

## 5. Conclusions

We performed genome-wide analysis and bioinformatic scanning of transposable elements of *Rhus* gall aphid *Schlechtendalia chinensis*, providing in-depth information about the evolution of TEs in the *S. chinensis* genome. The high diversity and presence of transcriptionally active TEs in the genome showed different waves of TE invasion and activities of TEs in the *S. chinensis* genome. Transposase-derived proteins in *S. chinensis* related to higher vertebrates provide evidence of their ancient domestication and high diversity. A large number of TE insertions in the functional genes provide insight into the insertional inactivation of genes, which could play a crucial role in genomic adaptations. This study provides detailed information about TEs in the *S. chinensis* genome. It could be used as a reference database for comparative genomic analysis of TEs between galling and non-galling aphids, and also afford a reference for genomic evolutionary research.

## Figures and Tables

**Figure 1 ijms-23-15967-f001:**
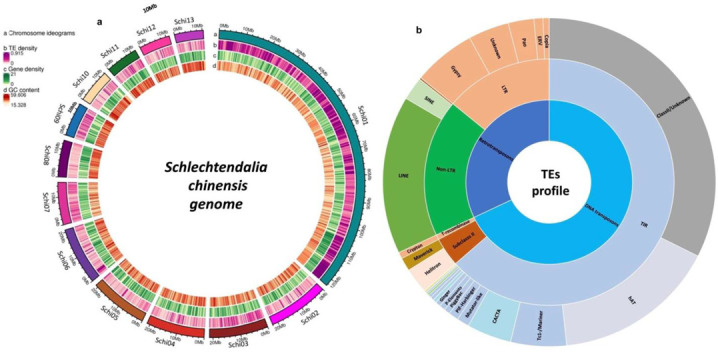
TEs’ distribution in the genome of *Schlechtendalia chinensis*. (**a**) Showing TE density in the 13 chromosomes of *S. chinensis*. (**b**) Showing TE classification and contribution to the genome size.

**Figure 2 ijms-23-15967-f002:**
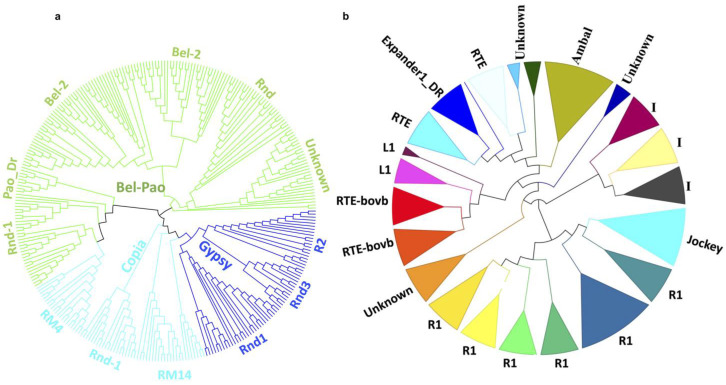
Phylogenetic classification of LTR and non-LTRs retrotransposons. (**a**) Classification of the 309 full-length representative sequences from the three major superfamilies, i.e., Copia, Gypsy, and Pao, into known clades. (**b**) Classification of 219 full-length representative sequences of the LINE superfamily into different known lineages and three unknown lineages.

**Figure 3 ijms-23-15967-f003:**
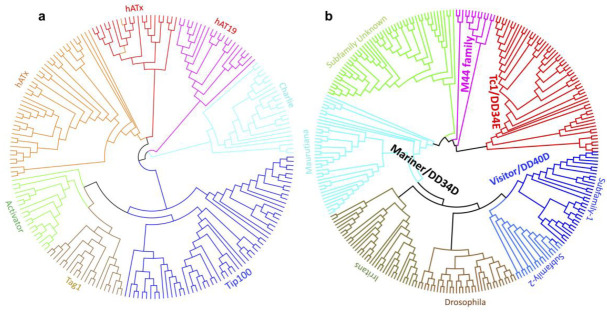
Phylogenetic classification of the two major superfamilies of Class II TEs in the genome of *S. chinensis*. (**a**) Classification of 207 full-length representative sequences from hAT superfamily, classifying them into six known families. (**b**) Classification of 287 full-length representative sequences of the Tc1/mariner superfamily into four prominent families and one unknown clade. The *Mariner* family is divided into three subfamilies using known reference sequences from GenBank. DD40D/Visitor is identified for the first time in aphids in this study. The M44 is the new uncharacterized family, and sequences in this analysis were placed in this family following the Dfam.org classification.

**Figure 4 ijms-23-15967-f004:**
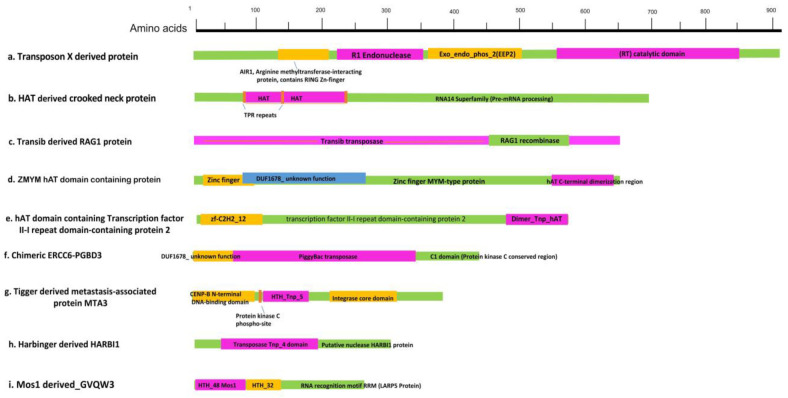
Transposon-derived protein sequences (domesticated) in the genome of *S. chinensis*. Amino acid sequences analysis of the eight (a–i) transposon-derived proteins retrieved representing at least one transposon domain domesticated in them. The sequence regions colored pink and orange in the sequences are mainly derived from TEs. The scale at the top represents the number and position of amino acids in the proteins.

**Figure 5 ijms-23-15967-f005:**
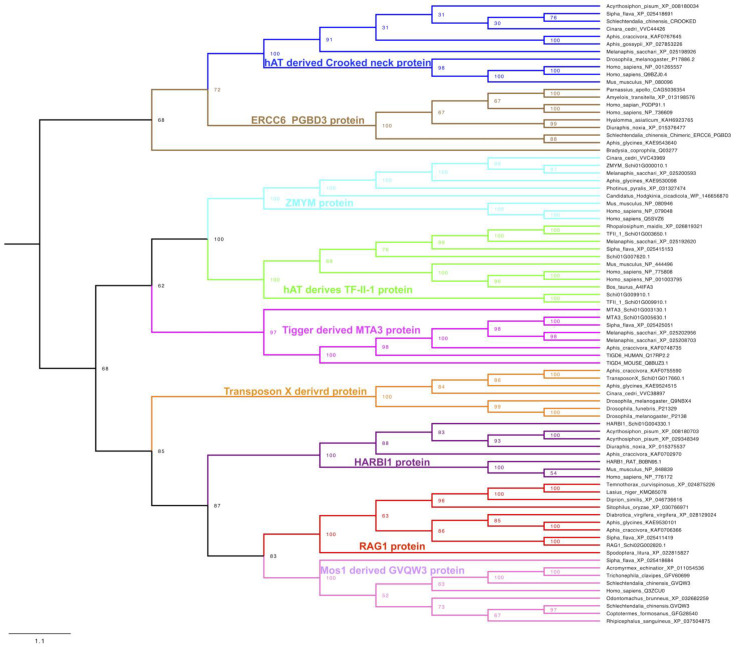
Phylogeny of the transposon-derived proteins retrieved from the genome of *S. chinensis* with other organisms retrieved from NCBI GenBank as a result of BlastP. Phylogenetic analysis of the sequences indicates their ancient domestications and can be found in insects to higher vertebrates. Clustering of the sequences derived from vertebrates, including humans with *S. chinenis* and other insects in the tree, represent their common origin. Each branch color in the tree represents a different group of transposon-derived proteins originating from the domestication of different TEs.

**Figure 6 ijms-23-15967-f006:**
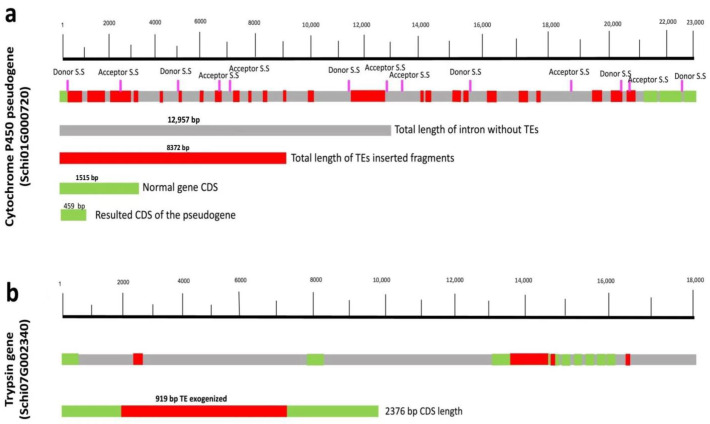
Graphical representation of TE insertion in the intron and exon of the functional gene in *S. chinensis*. (**a**) TE insertions (labeled red) in the intron (labeled grey) of the Cytochrome P450 pseudogene (SchiG000720). TE insertions contain several splicing sites, Donor and Acceptor (labeled pink), that could have led to the inactivation of the gene. The 1515 bp exon regions are labeled green. The scale at the top indicates the nucleotides’ position and sequence length. (**b**) TE insertions (labeled red) in the functional gene Trypsin (SchiG002340) show a 919 bp long TE fragment inserted in the exon of the genes without interrupting the coding activity, indicating its exogenization.

**Figure 7 ijms-23-15967-f007:**
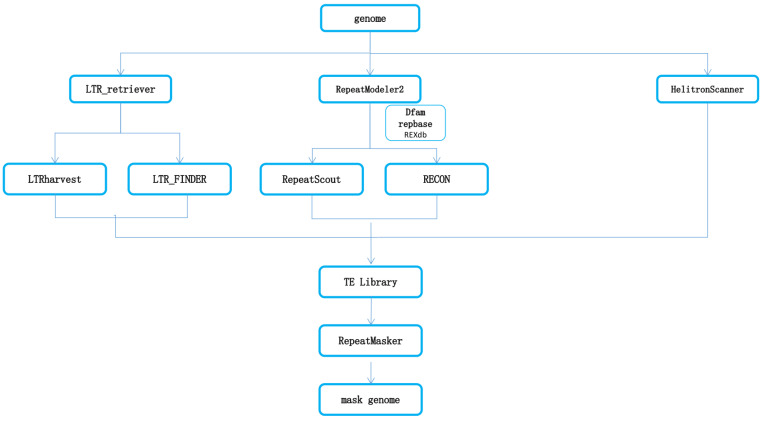
Schematic representation of TEs mining, identification, annotation, and classification using a collection of bioinformatics tools.

**Table 1 ijms-23-15967-t001:** Classification and proportion of transposable elements identified in the genome of *Schlechtendalia chinensis*.

Class	Order	Superfamily	No. of Copies	Full-Length Copies	Total Length of All Copies	Percentage in Genome
Retrotransposons	LTR	Copia	1796	100	463,436	0.13
		Gypsy	18,239	2002	10,381,315	3.01
		ERV	7463	0	513,028	0.15
		Pao	2624	179	1,047,169	0.3
		Cassandra	6	0	336	0
		Caulimovirus	3	0	171	0
		Unknown	11,892	220	3,023,472	0.88
	Non-LTR	DIRS	481	0	26,185	0.01
		Viper	52	0	5262	0
		LINE	46,644	276	16,342,622	4.74
		SINE	6852	-	1,234,961	0.36
	Subtotal		96,053	2777	33,038,958	9.59
DNA transposons	TIR	Tc1-/Mariner	16,200	571	3,923,428	1.14
		CACTA	21,973	53	2,506,904	0.73
		hAT	47,622	990	12,144,219	3.52
		Mutator-like	3610	0	921,870	0.27
		PIF-Harbinger	2929	11	458,516	0.13
		PiggyBac	1774	37	415,325	0.12
		P elements	1513	09	382,695	0.11
		Ginger	1725	0	235,990	0.07
		Zisupton	1081	0	62,632	0.02
		Novosib	969	01	81,878	0.02
		Kolobok	897	0	56,957	0.02
		Sola	508	0	70,640	0.02
		IS3EU	484	0	33,525	0.01
		Merlin	343	07	69,223	0.02
		Dada	332	0	28,627	0.01
		Zator	68	0	4652	0
		Academ	58	0	4019	0
	Tyrosine-recombinase	Crypton	1929	0	244,861	0.07
Class II (Subclass 2)						
	Helitron	-	7503	158	2,623,664	0.76
	Maverick	-	3899	218	4,729,237	1.37
	Class II/Unknown		97,190	2911	27,695,009	8.04
	Subtotal		212,638	4966		
Total			308,196	7743	89,735,022	26.04

**Table 2 ijms-23-15967-t002:** Detailed information of the transcriptionally active transposable elements retrieved from the transcriptome of *Schlechtendalia chinensis*.

Class	Superfamily	No. of Transcripts	Shortest Sequence Length	Longest Sequence Length	Average Length
Retrotransposons					
	Pao	10	103	1732	879.5
	Copia	2	25	222	148.5
DNA-transposons					
	Mariner	54	51	416	112.4
	hAT	8	54	600	316
	PiggyBac	22	56	508	128
	Mutator-like	4	156	648	454
Sub-class 2					
	Helitron	9	55	134	106

**Table 3 ijms-23-15967-t003:** Detailed information of the three families of functional genes, and transposable elements’ insertions in them.

	Cytochrome P450	Carboxyesterase	Trypsin
Number of genes	36	21	44
Total length of genes (bp)	265,062	244,753	309,547
Total number of exons	284	176	320
Average number of exons per gene	7.88	8.8	7.27
Average length of exons per gene (bp)	1461.25	1771.40	1529
Total length of exons (bp)	52,605	35,428	67,281
Total number of introns	179	156	278
Average length of introns per gene (bp)	5901	10,466.25	5506
Total length of introns (bp)	212,457	209,325	242,266
Total number of TE insertions (≥100 bp)	186	195	151
Total length of TE fragments (bp)	31,774	33,497	25,710
Number of TE insertions in introns	180	192	133
TE fragments in CDS	6	3	18
Number of pseudogenes	1	0	0

## Data Availability

High-throughput sequencing data analyzed in this project and the whole-genome project (including assembly and annotation) are deposited under BioProject (PRJNA833747) and BioSample (SAMN28016330) at NCBI GenBank. The accession number for each chromosome will be updated later when received from NCBI. All other data generated during this study are included in this article and its Appendix A. The chromosome-level genome assembly will also be provided upon request to the corresponding author if not yet updated on NCBI.

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
