# Peer review of "Mobilome of the Rhus Gall Aphid Schlechtendalia chinensis Provides Insight into TE Insertion-Related Inactivation of Functional Genes"

_ijms, 2022, doi:10.3390/ijms232415967_

Round 1

Reviewer 1 Report

The authors analyze the gall aphid Rhus Schlechtendalia chinensis mobiloma, highlighting a number of interesting data. Although the topic of the article is interesting, I also consider it to be 'immature' and it would be appropriate to make some changes before publication.

Below are some suggestions for authors.

Line 58-70: in order to the insertional mutagenesis cite Palazzo et al. (doi: 10.3390/cells11030583).

There is a different reference formatting (i.e. lines 399 (Scalzitti, et al. 2021) or  572: Jurka et al. 2005)

In the introduction the cited literature is not always suitable (see reference 16-18).

Among the results, the authors should mention some key information about the genome and its organization. For example they do not say how many Mb the genome is and how it is organized but, they report data in percentages (“… DNA transposons, accounting for approximately 7% of the 101 whole genome”).       

Line 125: The author writes: “LTR-RTs covered 42,638,927 bp, accounting for 4.48%”. I think that the number is wrong.

Figure 2 is unclear. What is meant by Rnd1 (sometimes write as Rnd-1), Rnd3, RM4.

Table 1. it would be more informative if the authors added columns with the number of potentially active elements and the full-length TE.

Line 288: the authors perform a transcriptional analysis but, in the file S6, are reported only aminoamides sequence… what does that means?

Figure 4. line 300: “… the Y-axis represents each transcript's length”. I understood that the longest transcript is 1800 bp. if this is correct, I think that for PiggyBac, for example, the longest transcript is almost 600 bp long. but this means that the longest transcript is only part of PiggyBac's transposase. Is this correct? I suggest implementing the figure with the expected length of the transcript and clarify this passage in the main text.

Line 383: the organization of cytochrome P450 is very interesting (in Drosophila, the transposon Bari1 has a similar effect on the same gene; see review doi: 10.3390/cells11030583). The authors said: 'A total of 186 TE fragments' which TE families are involved? What role do these TE fragments play? Have the authors done any further analysis on the expression profile of these genes?

Line 490-501: cite the review: doi: 10.1080/07388551.2021.1888067

Line 510: With reference 46, cite doi: 10.1016/j.gene.2005.06.005

There are a lot of repetitive sentences (see line 74-75);

Author Response

Dear Reviewer, thank you so much for your time and valuable suggestions. Please find below (attached) point by point response to each suggestions.

Sincerely

Reviewer 2 Report

The authors presented a comprehensive bioinformatic and functional characterization of transposable elements (TEs) in the sequenced genome and transcriptome of the aphid Rhus gall Schlechtendalia chinensis. They found very interesting cases of TE integration into genes, some of which are harmless due to inframer or intron insertion, while others lead to the synthesis of shortened proteins. Undoubtedly, transposition events into protein-coding regions affect protein function and contribute to insect evolution.

I have a few minor comments.

Have you detected TE insertions in the 3'-untranslated region (UTR) or 5'-UTR genes?

It would be good to provide more detailed statistics on TE site insertions (including introns, or maybe 3'-UTR, 5'-UTR) and the functional consequences of TE insertions in Table 3.

Figure 7a. There seems to be a typo in the length of the normal CDS; it should be 1.515 bp instead of 151.5.

Author Response

Dear Reviewer, Thank you so much your time and valuable suggestions. Below is attached the point by point response to each comment. 

Sincerely

Round 2

Reviewer 1 Report

Figure 2 is unclear. What is meant by Rnd1 (sometimes write as Rnd-1), Rnd3, RM4.

Response: The initial classification at superfamilies level was carried our using Repeatmodeler2 for automatic identification and classification of TEs families. The repeatmodeler2 further classify the families in to different lineages with automated naming i.e., Rnd-1, RnD-2, RM4 etc. We performed phylogenetic and classify the lineages in to subfamilies based on Dfam and Repbase classification. Lineages having no subfamily information in the two databases, were named per repeatmodeler classification. See classification in Supplemenatry data 2 and 3. Figure 2 was also revised and the names Rnd-1, Rnd1 were revised and corrected.

Please add a reference about this classification

Table 1. it would be more informative if the authors added columns with the number of potentially active elements and the full-length TE.

Response: Thanks for the suggestion. We have revised and add information about full length copies, in the Table 1, while the potentially active copies were found in only few groups, i.e., Mariner, hAT etc, and is already discussed in result and given in supplementary data where required.

The full-length data reported in table 1 are not informative. This information should be available  for each single superfamily.

Line 288: the authors perform a transcriptional analysis but, in the file S6, are reported only aminoamides sequence... what does that means?

Resposne: We carried out genome assembly and transcriptomic analysis for the given species, followed by TEs mining and classification, which was main target of the present study. As having transcriptomic data in hand, we analyzed the transcript of S.chinensis to provide basic(no detailed) information about TEs expression in the genome. The supplementary data contains the translated amino acid sequences of the TEs transposases of S.chinensis.

I disagree with the authors because an overview of TE expression is very different by a protein analysis. If the aim was to report the transcripts found, then the authors must report the transcripts and not the aminoacids. What they do is an exercise very hard, because in literature are reported a lot of cases in which TE expression analysis, where transcripts and proteins detected don't mach. Furthermore how they said: “…we analyzed the transcript of S.chinensis to provide basic(no detailed) information about TEs expression in the genome.”, the expression profile produces a list of transcripts.

Figure 4. line 300: “... the Y-axis represents each transcript's length”. I understood that the longest transcript is 1800 bp. if this is correct, I think that for PiggyBac, for example, the longest transcript is almost 600 bp long. but this means that the longest transcript is only part of PiggyBac's transposase. Is this correct? I suggest implementing the figure with the expected length of the transcript and clarify this passage in the main text.

Response: Yes we have just presented the transcripts of active TEs, or transcripts having signature of TEs in the S. chinensis. As almost all the transcripts have shorter length as expected that could be due immature stop codons, as prevalent in TEs or activity of exonucleases after expression. It is hard to have a clear statement, without further experimental validation, but the

present figure provide a basic map of active TEs that could be targeted for experimental expression and transposition analysis in the future studies.

Without the number of active elements (which should be included in table 1), this figure is not useful. However, leaving this aside, I consider the representation of this figure to be very misleading. The authors write of active elements to which a full-length TE transcript does not correspond. In my opinion it is a result a little be forced. I suggest removing this data or rewrite the paragraph with more supported conclusions.

Author Response

Dear Reviewer

Thanks for your time and suggestions. Please find below (attachment)point by point response to each comment.

Round 3

Reviewer 1 Report

In my opinion, the paper is ready for publication.